# A Method for Spatiotemporally Merging Multi-Source Precipitation Based on Deep Learning

**Wei Fang** [1,2] , **Hui Qin** [1,2,*] , **Guanjun Liu** [1,2] , **Xin Yang** [1,2] , **Zhanxing Xu** [1,2,3] , **Benjun Jia** [4] **and Qianyi Zhang** [1,2]

1   School of Civil and Hydraulic Engineering, Huazhong University of Science and Technology, Wuhan 430074, China; d202081169@hust.edu.cn (W.F.); d202081167@hust.edu.cn (G.L.); d201880948@hust.edu.cn (X.Y.); zhanxing@hust.edu.cn (Z.X.)
2   Hubei Provincial Key Laboratory of Digital Watershed Science and Technology, Huazhong University of Science and Technology, Wuhan 430074, China
3   Power China Huadong Engineering Corporation Limited, Hangzhou 311122, China
4   Hubei Key Laboratory of Intelligent Yangtze and Hydroelectric Science, China Yangtze Power Co., Ltd., Yichang 443000, China
*   Correspondence: hqin@hust.edu.cn; Tel.: +86-186-7408-4718

**Abstract:** Reliable precipitation data are essential for studying water cycle patterns and climate change. However, there are always temporal or spatial errors in precipitation data from various sources. Most precipitation fusion methods are influenced by high-dimensional input features and do not make good use of the spatial correlation between precipitation and environmental variables. Thus, this study proposed a novel multi-source precipitation spatiotemporal fusion method for improving the spatiotemporal accuracy of precipitation. Specifically, the attention mechanism was used to first select critical input information to dimensionalize the inputs, and the Convolutional long-short-term memory network (ConvLSTM) was used to merge precipitation products and environmental variables spatiotemporally. The Yalong River in the southeastern part of the Tibetan Plateau was used as the case study area. The results show that: (1) Compared with the original precipitation products (IMERG, ERA5 and CHIRPS), the proposed method has optimal accuracy and good robustness, and its correlation coefficient (*CC*) reaches 0.853, its root mean square coefficient (*RMSE*) decreases to 3.53 mm/d and its mean absolute error (*MAE*) decreases to 1.33 mm/d. (2) The proposed method can reduce errors under different precipitation intensities and greatly improve the detection capability for strong precipitation. (3) The merged precipitation generated by the proposed method can be used to describe the rainfall–runoff relationship and has good applicability. The proposed method may greatly improve the spatiotemporal accuracy of precipitation in complex terrain areas, which is important for scientific management and the allocation of water resources.

**Keywords:** A-ConvLSTM model; multi-source precipitation products; spatiotemporal fusion method; deep learning; precipitation accuracy improvement

## 1. Introduction

Precipitation is the key climatic factor for describing the hydrological cycle system in a region [1,2]. High-precision precipitation estimates can effectively reflect the surface environmental conditions and the spatiotemporal distribution of a water cycle, which is helpful to provide the scientific basis for decision making for water resource management [3]. Furthermore, precipitation is also an important indicator of climate change [1,4]. In recent years, drought and flood disasters caused by too much or too little precipitation have occurred frequently because of global warming [5,6]. For example, in the Yangtze River Basin, severe floods displaced millions of residents due to extreme precipitation events in 2020 [7,8], while extreme drought caused difficulties in water supply and power consumption for millions of people due to the persistent lack of precipitation in the summer of 2022 [9]. These natural disasters caused by precipitation have seriously threatened

the safety of human life and property and sustainable economic development. Therefore, high-precision precipitation with high spatiotemporal resolution is very important for the study of water cycles, the reduction of drought and flood disasters and the sustainable development of the economy, etc.

At present, the sources of precipitation data are mainly ground observation data, radar monitoring data, satellite retrieval data and reanalysis products [3,10]. Ground observation precipitation is directly measured by the rain gauge, so its accuracy is generally considered to be the highest [11,12]. Because they are affected by topographic conditions and economic development, the number and density of ground monitoring stations are different in different places [13,14]. As for China, the east has a much higher density of ground monitoring stations than the west, and plain areas have a much higher density than mountain and plateau areas [12]. Thus, although ground observation precipitation has high accuracy, it cannot reflect the spatial characteristics of precipitation well, especially in high-altitude areas such as the Tibetan Plateau. Radar monitoring of precipitation, which is estimated from radar echo intensity, has good accuracy and can capture the spatial distribution information of precipitation over a large range [15,16]. However, radar precipitation still has a larger error than ground observation precipitation because radar monitoring is easily affected by various factors such as the terrain and climate [15,17]. Moreover, the construction cost of radar monitoring networks is high [3,15]. Thus, radar precipitation can only be used in a limited range and cannot meet the application requirements of complex terrain, such as the source region of the Yangtze River. Compared to the former two, precipitation products based on satellite and reanalysis are widely used in hydrological research because of their high spatiotemporal resolution [16–18]. For example, the Integrated Multi-satellite Retrievals for GPM (IMERG) can provide precipitation with a 0.1° spatial and 30-min temporal resolution, and Climate Hazards Group InfraRed Precipitation with Station Data (CHIRPS) can provide precipitation with a spatial resolution of 0.05° [18–20]. Although the satellite retrieval of precipitation is not constrained by geographical conditions, it still has large errors due to the influence of retrieval algorithms and sensors [19,21]. To improve the accuracy of observations, reanalysis products combining numerical model data with various observations are produced by quality control and data assimilation algorithms [4,21–23]. They have been used as approximations of observations because of the advantages of high spatiotemporal resolution and long time series [21]. One of the well-known reanalysis products, ERA5, has provided 0.1° hourly precipitation data since 1950 [24,25]. However, because they are affected by numerical models and assimilation algorithms, reanalysis products inevitably contain systematic and random errors which call for further improvement [18,23]. Therefore, constrained by geographical environments, precipitation measurement's high cost, numerical models and algorithms, multi-source precipitation cannot be directly applicable to hydrological research in a certain basin.

Now, it has become a mainstream way to study multi-source precipitation fusion methods for improving precipitation accuracy [26–29]. The basic idea is that using ground observation precipitation as the benchmark, high-precision precipitation datasets with long sequences are obtained by merging radar precipitation, satellite precipitation, reanalysis products and other auxiliary variables with ground observation precipitation, based on fusion models [3,17]. Traditional mathematical regression models (such as mean bias correction, geographically weighted regression, optimum interpolation and so on) were used for global or local correction of multi-source precipitation early on [30–33]. For example, Chao et al. [32] used the geographically weighted regression model to merge ground observation precipitation and CMORPH precipitation data, improving the accuracy and applicability of CMORPH. Shen et al. [17] proposed a new fusion model based on local gauge correction and optimal interpolation to produce China's 1-km gauge–radar–satellite-merged precipitation dataset. However, traditional mathematical regression models rely too much on strong assumptions to adequately capture the non-linear relationship between ground observation precipitation and other precipitation products [12,34]. Instead, machine learning models have been introduced for multi-source precipitation fusion methods

due to their strong nonlinear learning ability, including neural networks, support vector machine and tree models [35–37]. For example, Zhang et al. [36] proposed a novel double machine learning approach that could better capture the temporal dynamics of precipitation, realizing the fusion of multiple satellite precipitation products and gauge observations. Zhang et al. [38] used the eXtreme Gradient Boosting model and the Kriging interpolation model to merge ground observation precipitation and radar precipitation, respectively, and indicated that the eXtreme Gradient Boosting model had a better effect.

In addition, it is of great significance for improving precipitation accuracy to consider the impact of environmental variables on precipitation. Many studies have shown that precipitation change is closely related to surface temperature and wind speed [39–41]. Fang et al. pointed out that topographic factors (such as slope, aspect and terrain roughness) have important effects on precipitation [42]. Jia et al. developed the multi-source precipitation fusion study in Qaidam Basin and indicated that the addition of a normalized difference vegetation index (NDVI), elevation and aspect could promote the improvement of precipitation accuracy [43]. Therefore, various environmental variables have been added as fusion models' driving inputs. For example, Hong et al. [44] added elevation, surface pressure and wind speed to the input factors to merge gauge, satellite and reanalysis data based on artificial neural networks and developed a daily precipitation dataset with a spatial resolution of 0.1° in the Tibetan Plateau. Jing et al. [45] introduced surface temperature as one of the Random Forest inputs to downscale the yearly TRMM 3B43 V7 precipitation data and improve the temporal and spatial accuracy of precipitation. Furthermore, other environmental variables, including longitude, latitude and slope, can also contribute to the improvement of precipitation accuracy [46,47].

Although great progress has been made in the study of multi-source precipitation fusion in recent decades, most fusion models based on traditional mathematical regression models or general machine learning models only focus on one aspect of the temporal–spatial correlation between ground observation precipitation and other precipitation products rather than both concurrently [12]. For example, the spatial interpolation method can only reflect spatial correlation, while the support vector machine can only reflect temporal correlation. With the development of computer technology, deep learning methods, which have greater advantages in processing massive and spatiotemporal information, have been widely used in image recognition and video processing as well as streamflow and precipitation forecasting [48–51]. For example, Shi et al. [50] used convolutional long-short-term memory network (ConvLSTM) for precipitation forecasting and indicated that ConvLSTM could reflect spatiotemporal correlations well. Liu et al. [52] proposed a directed graph deep neural network that could effectively capture the spatiotemporal information of precipitation and streamflow and achieved good results in streamflow forecasting. It can be seen that the emergence of deep learning models can provide the possibility to handle the temporal–spatial nonlinear relationship between ground-observed precipitation and other precipitation products better. However, to the best of our knowledge, spatiotemporal deep learning methods have been less applied in the study of multi-source precipitation fusion. In addition, most studies were developed in areas with dense precipitation stations and seldom evaluated the practicability of the results from the perspective of the rainfall–runoff relationship well [1,10,12,26].

In this study, we aim to propose a multi-source precipitation fusion model based on deep learning to improve the spatiotemporal accuracy of precipitation estimates in the Yalong River. The purposes of this study are as follows: (1) to propose a multi-source precipitation spatiotemporal fusion model by coupling the attention mechanism and ConvLSTM; (2) to verify the effectiveness of the proposed model through comparing other models and evaluating its performance under different precipitation intensities; (3) to produce a multi-source merged precipitation dataset and evaluate its availability using hydrological models. The remainder of this study is organized as follows: The details of the study area and data and methods are given in Section 2. The experimental results and discussion are presented in Section 3. Finally, conclusions are drawn in Section 4.

## 2. Data and Methodology

### 2.1. Study Area and Data

2.1.1. Study Area

In this study, the Yalong River was used as the research area to verify the effectiveness and applicability of the proposed method. Located in the southern part of the Tibetan Plateau, the Yalong River (Figure 1) is the largest tributary of the Jinsha River in the upper reaches of the Yangtze River, which is also a typical high mountain and canyon river. It has abundant hydropower resources, with a total length of 1571 km and a drainage area of about 136,000 km². There are five important hydrological stations from upstream to downstream in the basin, namely Ganzi, Yajiang, Maidilong, Jinping and Tongzilin. The altitude of the Yalong River is mostly over 1500 m, and the terrain is higher in the northwest and lower in the southeast, with a huge natural precipitation [53,54]. Affected by topographic conditions and monsoon climate, the precipitation is unevenly distributed in time and space. In terms of space, the annual average precipitation increases from northwest to southeast and varies greatly from upstream to downstream; it is 600–800 mm upstream, 1000–1400 mm in the middle and 900–1300 mm downstream [55]. In terms of time, precipitation is mainly concentrated from June to September, and heavy precipitation is prone to occur in July and August [1,56]. However, it can be seen from Figure 1 that the precipitation stations in the Yalong River are relatively scarce and mainly concentrated in low-altitude areas due to the constraints of terrain conditions and the economic level. Therefore, to improve the spatiotemporal accuracy of precipitation in the Yalong River, it is of great significance to develop this multi-source precipitation fusion study.

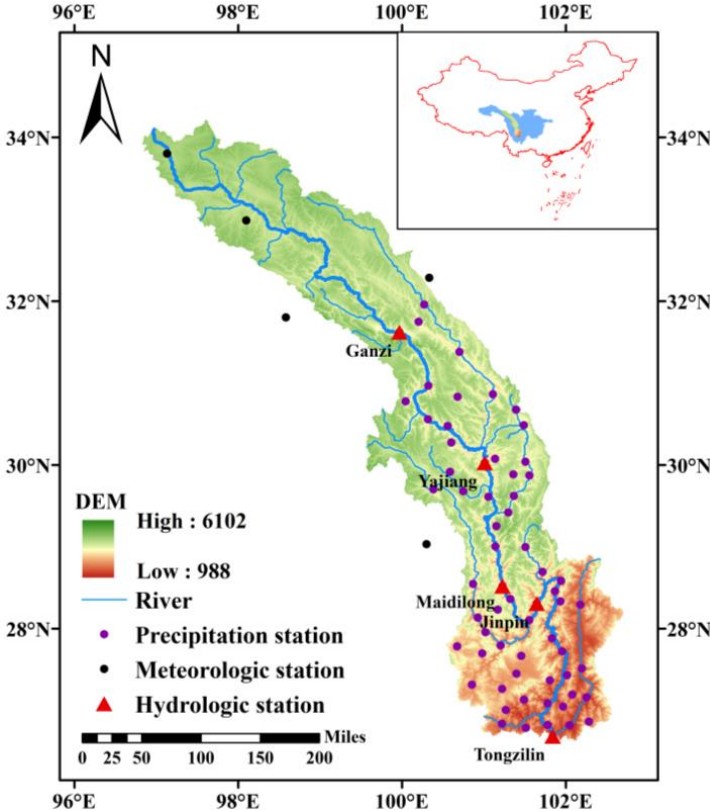

**Figure 1.** Terrain and station distribution in the Yalong River.

2.1.2. Data

Based on previous studies [1,2], ground observation precipitation, satellite precipitation data, reanalysis precipitation data, normalized difference vegetation index (NDVI), digital elevation model (DEM) and streamflow series of 5 hydrological stations from 2011 to 2020 were selected in this study. The basic information on the data used is listed in Table 1.

**Table 1.** Basic information of data used.

| Name | Spatial Resolution | Temporal Resolution | Source |
|---|---|---|---|
| Precipitation from precipitation station | \ | 1 d | Yalong River Hydropower Development Co., Ltd., China |
| Precipitation from meteorologic station | \ | 1 d | http://data.cma.cn/data/cdcdetail/dataCode/SURF_CLI_CHN_MUL_DAY_V3.0 (accessed on 1 July 2023) |
| Streamflow from hydrological station | \ | 1 d | Yalong River Hydropower Development Co., Ltd., China |
| IMERG | 0.1° | 1 d | https://disc.gsfc.nasa.gov/datasets/GPM_3IMERGDF_06/summary (accessed on 1 July 2023) |
| CHIRPS | 0.05° | 1 d | https://data.chc.ucsb.edu/products/CHIRPS-2.0/ (accessed on 1 July 2023) |
| ERA5 | 0.1° | 1 h | https://cds.climate.copernicus.eu/cdsapp#!/dataset/reanalysis-era5-land?tab=form (accessed on 1 July 2023) |
| NDVI | 0.05° | 16 d | https://search.earthdata.nasa.gov/ (accessed on 1 July 2023) |
| DEM | 90 m | \ | http://www.gscloud.cn/sources/accessdata/305?pid=302 (accessed on 1 July 2023) |

(1) Ground observation precipitation

In this study, ground observation precipitation was obtained from the real-time monitoring data of 67 precipitation stations and 5 meteorological stations. Precipitation stations were set up by the Yalong River Hydropower Development Co., Ltd. to monitor the precipitation information, whose average control area was 1500 km$^2$. However, their distribution was uneven due to the influence of terrain conditions, and there were few stations above the Ganzi hydrological station (Figure 1).

To fully verify the applicability of the proposed model, meteorological stations were added to provide upstream precipitation information and were from the Chinese Surface Climate Data Daily Value dataset (V3.0) provided by the China Meteorological Data Service Center. This dataset's accuracy and completeness were improved after quality control and provided reliable long-sequence daily precipitation data. In this study, to avoid duplication in the location of precipitation stations, daily precipitation data on 5 meteorological stations around the Yalong River (Figure 1) between 2011 and 2020 were selected.

(2) IMERG

As a typical representative product of the Global Precipitation Measurement Mission (GPM), the Integrated Multi-satellite Retrievals for GPM (IMERG) can provide precipitation data with a maximum temporal resolution of 0.5 h and a spatial resolution of 0.1°. According to the calibration times of precipitation data, IMERG can be subdivided into "early run", "late run" and "final run". Among them, the IMERG final run is currently the most applicable because it is bias-corrected based on ground observation precipitation [57–60]. Recently, it has been updated to Version 06. Therefore, the GPM IMERG Final Precipitation V06 product, containing level 3 data between 2011 and 2020, was selected in this study.

(3) CHIRPS

Climate Hazards Group InfraRed Precipitation With Station Data (CHIRPS) is a precipitation product jointly created by the United States Geological Survey and the University of California for trend analysis and seasonal drought monitoring [61]. It has provided precipitation data with a spatial resolution of 0.05° since 1981, and its latest version is V2.0 [62]. In this study, we selected CHIRPS-2.0 daily data with a spatial resolution of 0.05° between 2011 and 2020.

(4) ERA5

ERA5 is the fifth generation of atmospheric reanalysis global climate data produced by the European Centre for Medium-Range Weather Forecasts (ECMWF). It was generated by merging multiple sources of observation data, based on the four-dimensional assimilation system and advanced parameterization schemes. Therefore, it has a very high spatiotemporal resolution and can provide hourly estimates of hydrometeorological variables such as temperature, wind, precipitation and so on. In this study, ERA5-Land hourly precipitation data with a spatial resolution of 0.1° between 2011 and 2020 were used. In addition, temperature and wind data from ERA5-Land were also selected as auxiliary variables of the fusion model.

(5) Other data

Normalized difference vegetation index (NDVI), which is an important indicator for the macroscopic monitoring of vegetation, can reflect vegetation coverage and vegetation growth. In general, NDVI has a positive correlation with precipitation [43,63]. In this study, we used MODIS 16-day NDVI data with a spatial resolution of 0.05° between 2011 and 2020.

The digital elevation model (DEM) is used to digitally express the shape of the terrain surface, which can reflect information on a region's topographic factors such as slope, aspect and elevation. In this study, the DEM data, with a spatial resolution of 90 m, were obtained from the Shuttle Radar Topography Mission system.

In addition, the daily streamflow series of hydrological stations including Ganzi, Yajiang, Maidilong, Jinping and Tongzilin (2011–2020) were used to explore the hydrological effects of the multi-source merged precipitation dataset in this study. They came from the monitoring data of hydrological stations built by the Yalong River Hydropower Development Co., Ltd. Interventionary studies involving animals or humans and other studies that require ethical approval must list the authority that provided approval and the corresponding ethical approval code.

### 2.2. Methods

In this study, the framework of the proposed multi-source precipitation spatiotemporal fusion method based on the attention mechanism and Convolutional LSTM Network (ConvLSTM), is shown in Figure 2. The steps of this method are as follows.

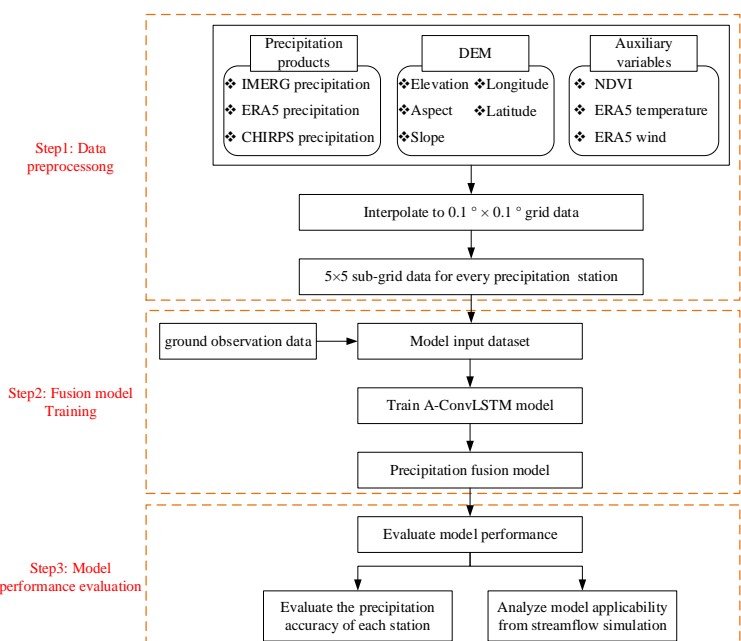

**Figure 2.** Flowchart of multi-source precipitation spatiotemporal fusion method based on attention mechanism and ConvLSTM.

(1) Data preprocessing: precipitation products, DEM and other auxiliary variables were collected and organized, which were unified into a daily scale in time and interpolated to 0.01° in space.

(2) Model building and training: The multi-source precipitation spatiotemporal fusion model was established by coupling the attention mechanism and ConvLSTM. The input dataset was constructed to train the fusion model by extracting the information on precipitation products, DEM and other auxiliary variables of the sub-grid around each precipitation station or meteorologic station.

(3) Model performance evaluation: According to the precipitation simulation results of each station, the accuracy of the fusion model was evaluated using evaluation indices. In addition, the multi-source fusion precipitation dataset of the entire study area, which was generated by the trained fusion model, was used to drive hydrological models to evaluate the practicability of the fusion model to the streamflow simulation results.

### 2.2.1. Data Preprocessing

There are missing and abnormal values in the ground precipitation data, precipitation product data, temperature data, wind data and NDVI data. Therefore, the missing values need to be processed to maintain data integrity. In this study, the missing and abnormal values of precipitation data were filled with zero, and other data were processed through linear interpolation.

Since the time scale of each data is different, it is necessary to unify the time scale. In this study, the time scale was unified into a daily scale according to the ground precipitation data. Therefore, the ERA5 daily precipitation data were obtained by accumulating, the ERA5 daily temperature and wind data were obtained by averaging, and the daily NDVI data were obtained through the strategy in which the daily values were set as equal to the 16-day values.

Since precipitation changes are related to terrain factors, factors such as elevation, slope and aspect need to be extracted from the DEM data, which were used as the input of the fusion model. In this study, the longitude, latitude, elevation, slope and aspect data of each sub-grid point were extracted from the DEM with the help of ArcGIS 10.2.

Since the spatial resolution of each data is initially different, the spatial resolution needs to be made consistent. In this study, the spatial resolution of the grid data was processed to 0.1° by the inverse distance weighting method.

By considering the variables related to precipitation in time and space, the input dataset of the fusion model was constructed. First, $n \times n$ sub-grids were extracted by taking each precipitation station or meteorologic station as the center. Second, the data of precipitation products, ERA5 temperature, ERA5 wind and NDVI were extracted corresponding to each simulation period in each sub-grid. Spatial features, including longitude, latitude, elevation, slope and aspect, were also extracted in each sub-grid. Finally, the model input dataset corresponding to the grid data and the ground observation precipitation in time and space was constructed. The detailed sub-grid extraction is shown in Figure 3.

### 2.2.2. Attention Mechanism

The attention mechanism, which is inspired by human visual attention, is proposed by researchers from Google [64,65]. In cognitive science, to rationally utilize limited visual information processing resources, focus is places on important parts of information while ignoring other visible information. Similarly to the human attention function, the attention mechanism in deep learning can focus on input factors that are more critical to output variables among many factors and reduce attention to other factors. The network structure of the attention mechanism is shown in Figure 4. The network mainly consists of three layers, namely the input layer, attention layer and output layer. First, the input layer receives the input factors. Second, the attention layer constructs the attention weight matrix based on the importance of the input factors to the decision variable to obtain the attention weights of each factor. Finally, the output layer outputs the weighted factors obtained

by multiplying the attention weights with the input factors. The general function of the attention mechanism is described as follows:

$$A = \text{softmax}(\boldsymbol{W} \times \boldsymbol{X} + b) \tag{1}$$

$$\boldsymbol{X}' = A \times \boldsymbol{X} \tag{2}$$

where $\boldsymbol{X}$ denotes the input vector, $\boldsymbol{W}$ and $b$ denote the weight vector and bias that need to be trained, $A$ denotes the attention weights and $\boldsymbol{X}'$ denotes the weighted vector.

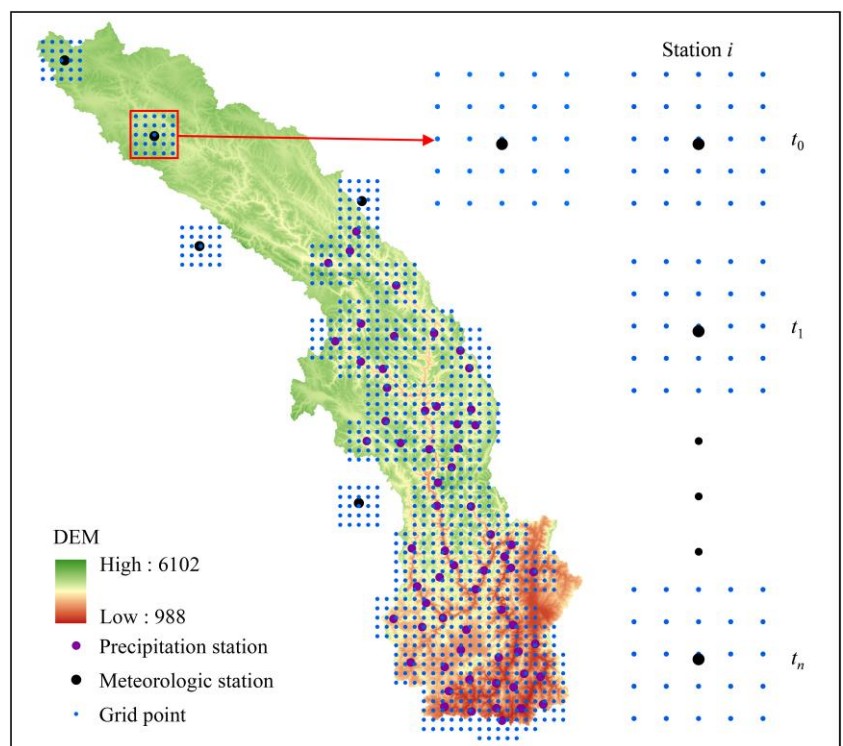

**Figure 3.** Diagram of sub-grid data extraction.

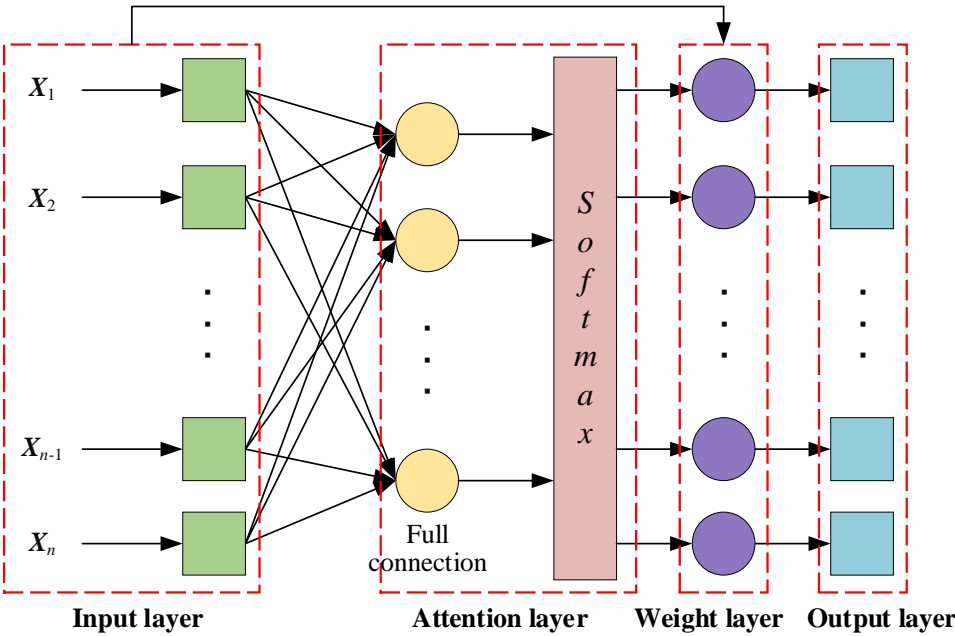

**Figure 4.** Network structure of attention mechanism.

### 2.2.3. ConvLSTM

As an improvement of the long-short-term memory network (LSTM), the Convolutional long-short-term memory network (ConvLSTM) not only has temporal modeling ability but can also describe local spatial features like a convolutional neural network [66]. LSTM consists of an input gate, forget gate, state gate and output gate, whose computation between network layers is processed similarly to a feed-forward neural network. Therefore, it can handle time series data well, but cannot describe the local spatial features well. To make up for this deficiency, ConvLSTM replaces the feed-forward connection between layers with convolutional structures, which is helpful for the extraction of spatial features. The internal detailed structure of ConvLSTM is shown in Figure 5, where the mathematical expressions are described as follows (other data were processed through linear interpolation):

$$I_t = \sigma(W_{XI} * X_t + W_{HI} * H_{t-1} + W_{CI} \circ C_{t-1} + b_I) \tag{3}$$

$$F_t = \sigma(W_{XF} * X_t + W_{HF} * H_{t-1} + W_{CF} \circ C_{t-1} + b_F) \tag{4}$$

$$\widetilde{C} = \tanh(W_{XC} * X_t + W_{HC} * H_{t-1} + b_C) \tag{5}$$

$$C_t = F_t \circ C_{t-1} + I_t \circ \widetilde{C}_t \tag{6}$$

$$O_t = \sigma(W_{XO} * X_t + W_{HO} * H_{t-1} + W_{CO} \circ C_t + b_O) \tag{7}$$

$$H_t = O_t \circ \tanh(C_t) \tag{8}$$

where $t$ denotes time, $*$ denotes convolution operation, $\circ$ denotes Hadamard product, $X$ denotes the input vector, $I$, $F$, $\widetilde{C}$, $C$, $O$ and $H$ denote the output of the input gate, forget gate, candidate memory, memory cell state, output gate and hidden layer, $W_*$ and $b_*$ denote the weight vector and bias of each gate or network layer that need to be trained, $\sigma(\cdot)$ denotes Sigmoid function and $\tanh(\cdot)$ denotes hyperbolic tangent function.

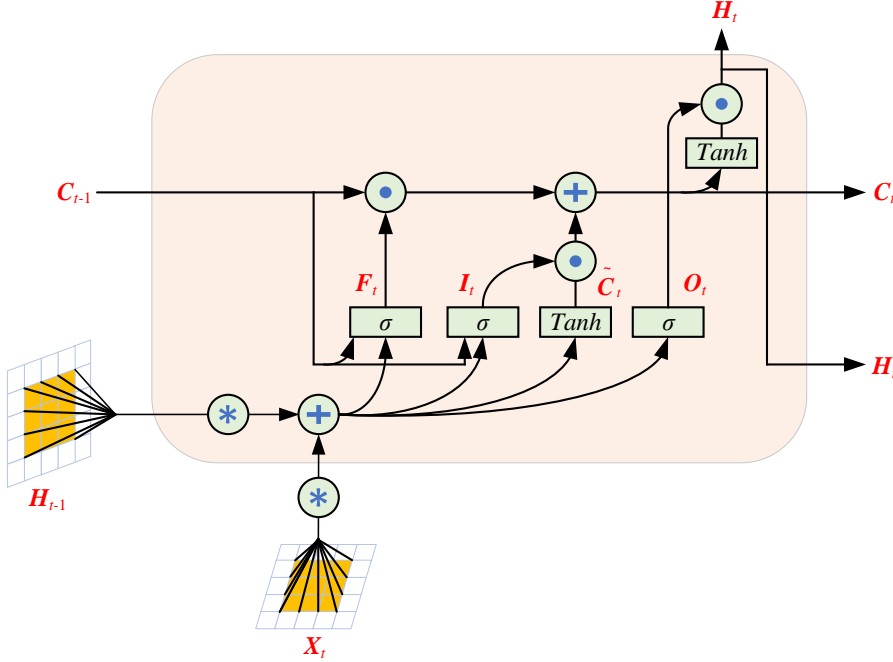

**Figure 5.** Internal detailed structure of ConvLSTM.

### 2.2.4. A-ConvLSTM

To accurately simulate spatiotemporal correlations between ground observation precipitation and other precipitation products, the multi-source precipitation spatiotemporal fusion model, by coupling the attention mechanism and ConvLSTM (A-ConvLSTM), is proposed to merge multi-source precipitation data. The attention mechanism is used to highlight the input factors that have greater impacts on the response variable, and ConvLSTM is used to capture spatiotemporal features. The structure of A-ConvLSTM is shown in Figure 6.

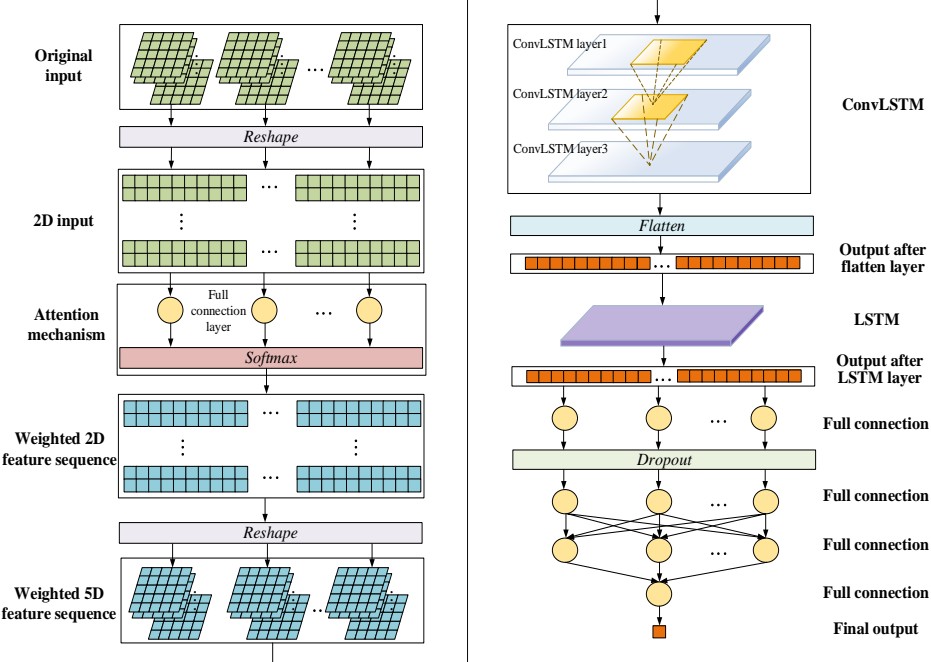

**Figure 6.** The structure of the A-ConvLSTM model.

In Figure 6, the squares represent different types of input factors, intermediate results and the output. The Reshape function is used to rescale the rows, columns and dimensions of the data to match the input type of the model without changing the characteristics of the original data. The attention mechanism, consisting of a fully connected layer and the Softmax function, is applied to highlight key input information. The 3-layer connected ConvLSTM layer is used to improve the spatiotemporal capture ability and merging accuracy of multi-source precipitation. The Flatten layer is used to one-dimensionally process the multi-dimensional output of the ConvLSTM network and then decode the output through the LSTM layer. Dropout is a regularization function to prevent overfitting, and dense is a fully connected layer. Therefore, the main coupling steps of A-ConvLSTM are as follows:

(1) The attention mechanism assigns attention weights to the two-dimensional inputs transferred from the original multi-dimensional inputs through the reshape function, and a weighted two-dimensional feature sequence is obtained.

(2) The reshape function converts weighted features into five-dimensional data as inputs of ConvLSTM to extract the spatiotemporal features of multi-source precipitation.

(3) The outputs of ConvLSTM are decoded by the LSTM network, and the final results are obtained via the fully connected network.

### 2.2.5. Other Models

(1) MLR

Multiple linear regression (MLR) is suitable for characterizing the correlation between multiple independent variables and a dependent variable.

(2) SVR

Support Vector Regression (SVR) is a well-known machine learning based on the principle of structural risk minimization. It works by defining a suitable kernel function to find the optimal regression plane that roughly satisfies all the data.

(3) LSTM

As a special type of Recurrent Neural Network (RNN), Long Short-Term Memory Neural Network (LSTM) is used to replace the units of RNN with memory units to improve long-term memory ability. Memory units are located in the hidden layer, including the forget gate, input gate and output gate. They can process the incoming information from the previous stage, discard some useless information and store valuable information to update the cell to obtain the best output result.

(4) GRU

The network structure of the Gated Recurrent Unit (GRU) is similar to LSTM. Unlike LSTM, its memory unit has only two gates: the update gate and reset gate. Furthermore, GRU uses hidden states to convey information. Therefore, the structure of GRU is simpler than LSTM.

(5) XAJ

The Xin'anjiang model (XAJ) is an aggregate hydrological model that can be used in humid and semi-humid areas. The model first divides the basin into a number of unit areas, then calculates the streamflow of each unit, based on the principle of runoff generation under saturated conditions, performs the calculation of catchment flow, and finally superimposes the sink flow of each unit as the forecast basin process of the whole basin.

(6) SWAT

The Soil and Water Assessment Tool (SWAT) is a distributed watershed hydrologic model based on GIS. Firstly, it divides multiple hydrologic response units according to a combination of soil type, land use and topography. Secondly, the water balance equation is utilized to simulate the process of a terrestrial hydrologic cycle. Third, the Muskingum method or variable storage coefficients are used for catchment calculations on the river. Finally, the hydrological processes in the watershed are simulated based on the characteristics of the river network.

*2.3. Evaluation Indices*

In this study, five evaluation indices, including root mean square error (*RMSE*), mean absolute error (*MAE*), correlation coefficient (*CC*), Nash-Sutcliffe model efficiency coefficient (*NSE*) and average relative error (*MRE*), were used to evaluate the accuracy and applicability of the proposed A-ConvLSTM model. *RMSE* reflects the degree to which the simulated sequence deviates from the observed sequence. *MAE* reflects the average error range of the simulated sequence, which can accurately reflect the actual error. *CC* reflects the linear correlation between the simulated and observed series. *NSE* reflects the overall agreement between simulated and observed sequences. *MRE* reflects the relative error between the simulated and observed sequence, which can better reflect the credibility of simulation errors. *CSI* reflects the success rate of precipitation products in detecting actual precipitation events. These are described as follows:

$$RMSE = \sqrt{\frac{\sum\limits_{i=1}^{n}(S_i - O_i)^2}{n}} \tag{9}$$

$$MAE = \frac{1}{n}\sum_{i=1}^{n}|S_i - O_i| \tag{10}$$

$$CC = \frac{\sum\limits_{i=1}^{n}(S_i - \overline{S}_i)(O_i - \overline{O}_i)}{\sqrt{\sum\limits_{i=1}^{n}(S_i - \overline{S}_i)^2(O_i - \overline{O}_i)^2}} \tag{11}$$

$$NSE = 1 - \frac{\sum\limits_{i=1}^{n} (O_i - S_i)^2}{\sum\limits_{i=1}^{n} (O_i - \overline{O_i})^2} \tag{12}$$

$$MRE = \frac{1}{n} \sum_{i=1}^{n} \frac{|S_i - O_i|}{O_i} \tag{13}$$

$$CSI = \frac{N_{11}}{N_{11} + N_{10} + N_{01}} \tag{14}$$

where $O$ denotes the observed sequence, $\overline{O}$ denotes the mean value of the observed sequence, $S$ denotes the simulated sequence, $i$ denotes the time interval and $n$ denotes sequence length. $N_{xy}$ is the number of observed precipitation events in category $x$ detected by the precipitation product in category y ($x, y$ = 0, 1; 0 means no precipitation and 1 means precipitation).

## 3. Results and Discussion

### 3.1. Precipitation Fusion Performance with Different Models

To evaluate the performance of the A-ConvLSTM model, the 5-fold cross-validation method was used to randomly divide the used modeling data into five parts according to the distribution of seventy-two ground stations, four of which were used for training and one for testing. Different model input datasets were obtained based on the different numbers of sub-grids for the model training. Each model used the Bayesian optimization algorithm to optimize the parameters, which was trained 50 times and iterated 100 times for each training. The optimal results of each model under different sub-grid numbers are shown in Figure 7.

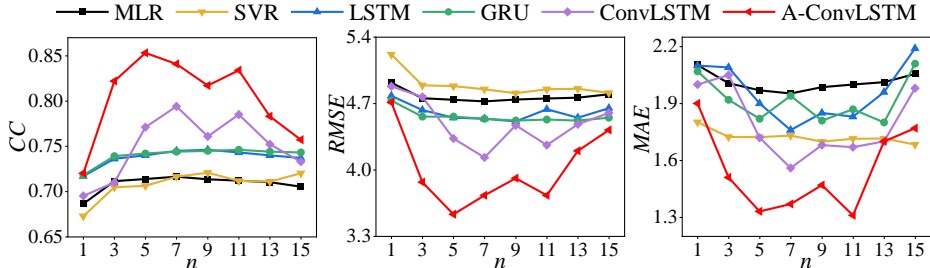

**Figure 7.** Optimal results of each model under different sub-grid numbers.

It can be seen from Figure 7 that as the sub-grid numbers increase, the accuracy of ConvLSTM and A-ConvLSTM first increase and then decrease, while the same trend is not very obvious in other models. This may be because ConvLSTM and A-ConvLSTM can capture the spatiotemporal feature of precipitation and can comprehensively merge sub-grid information. Within a certain range, the increase in the number of sub-grids, which means the addition of spatiotemporal information, can certainly improve the effect of precipitation fusion. However, increasing the number of sub-grids too much will cause too much useless information to interfere with the accuracy. Other models can only capture the temporal features of precipitation, and the addition of spatial information does not greatly improve their accuracy. Therefore, their accuracy varies relatively insignificantly with the number of sub-grids. Based on various evaluation indices, the sub-grid numbers selected for the optimal MLR, SVR, LSTM, GRU, ConvLSTM and A-ConvLSTM models were $7 \times 7, 9 \times 9, 9 \times 9, 11 \times 11, 7 \times 7$ and $5 \times 5$, respectively. In addition, the precipitation product data (IMERG, ERA5 and CHIRPS) of each ground station were obtained by the inverse distance interpolation method to be compared with the proposed A-ConvLSTM model. The performance of different precipitation fusion models is shown in Table 2.

**Table 2.** Accuracy of different precipitation fusion models.

| Model | CC | RMSE (mm/d) | MAE (mm/d) |
|---|---|---|---|
| IMERG | 0.452 | 6.11 | 2.64 |
| ERA5 | 0.390 | 7.01 | 3.83 |
| CHIRPS | 0.383 | 6.31 | 2.88 |
| MLR | 0.714 | 4.74 | 1.99 |
| SVR | 0.720 | 4.81 | 1.68 |
| LSTM | 0.746 | 4.51 | 1.85 |
| GRU | 0.747 | 4.53 | 1.87 |
| ConvLSTM | 0.791 | 4.17 | 1.71 |
| A-ConvLSTM | 0.853 | 3.53 | 1.33 |

It can be seen from Table 2 that, compared with the original precipitation products, the performance of precipitation fusion models is improved, and A-ConvLSTM performs best. In terms of evaluation indices, IMERG is the best and ERA5 is the worst, with *CCs*, *RMSEs* and *MAEs* of 0.452, 6.11 mm/d and 2.64 mm/d and 0.390, 7.01 mm/d and 3.83 mm/d, respectively. When MLR is used, the *CC* increases from 0.383 to 0.714, the *RMSE* decreases from 7.01 mm/d to 4.74 mm/d, and the *MAE* decreases from 3.83 mm/d to 1.99 mm/d, which indicates that considering the linear time-series correlation between input factors and the response variable can effectively improve the accuracy of merged precipitation. Compared with MLR, machine learning models such as SVR, LSTM and GRU have better precipitation fusion effects because of their strong nonlinear time-series fitting and generalization capabilities. The ConvLSTM model is clearly superior to the other models because it fully considers the spatial distribution of the input factors and integrates the terrain information around ground precipitation monitoring stations. However, because it is affected by irrelevant or weakly influential information in the input factors, the accuracy of ConvLSTM can still be further improved. A-ConvLSTM, which couples the attention mechanism and ConvLSTM, can provide a ConvLSTM with key information on the response variable in a large number of input factors to effectively improve accuracy and computational efficiency. The *CC* is increased from 0.383 to 0.853, and the *RMSE* and *MAE* are, respectively, 3.48 and 2.5 lower than for the original ERA5 data. Therefore, A-ConvLSTM has the best improvement in accuracy compared to the original precipitation products.

Based on the observed precipitation of ground stations, the *CC*, *RMSE* and *MAE* of each station are calculated for different precipitation fusion methods to obtain the spatial distribution of each index, as shown in Figures 8–10.

From Figures 8–10, it can be seen that the performance of each model is different at each ground station. Among the three precipitation products, IMERG performs best but has relatively large errors upstream and downstream of the Yalong River. Analyzing the possible factors, the following conclusions may be drawn: Firstly, due to the influence of the terrain, the precipitation process in high-altitude areas is usually more complex than that in plain areas, and there are fewer ground precipitation monitoring stations, which makes precipitation estimation by satellite sensors difficult [67–69]. Secondly, since precipitation in the Yalong River increases from northwest to southeast, high precipitation downstream increases the difficulty of satellite estimation [70]. ERA5 has the lowest accuracy, especially in the downstream areas, with large errors. This may be because the physical calculation process of ERA5 is mainly inclined toward high accuracy in Europe, and the high precipitation in the downstream of Yalong River increases the difficulty of its parameter adjustment [1]. CHIRPS has the smallest error fluctuation range among the three precipitation products, which indicates that its precipitation estimates are relatively stable at different stations. However, due to the limitations of a complex terrain and evolution algorithm [71,72], it still has large errors, especially downstream.

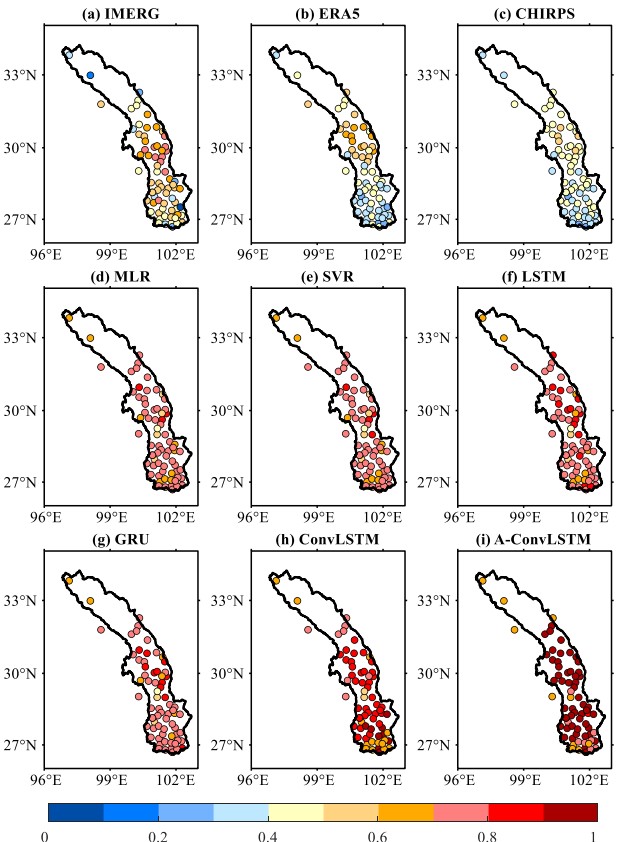

**Figure 8.** Correlation coefficient (*CC*) for different models.

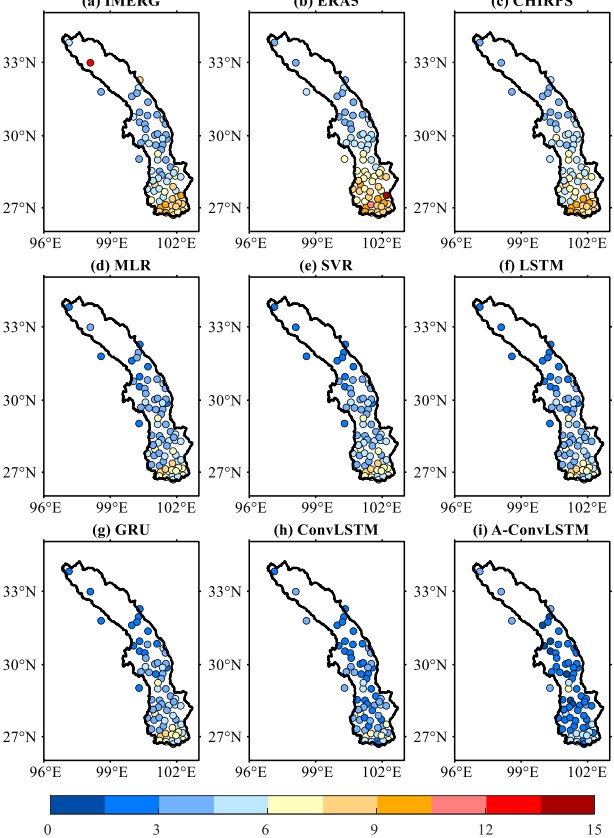

**Figure 9.** Root mean square error (*RMSE*) for different models.

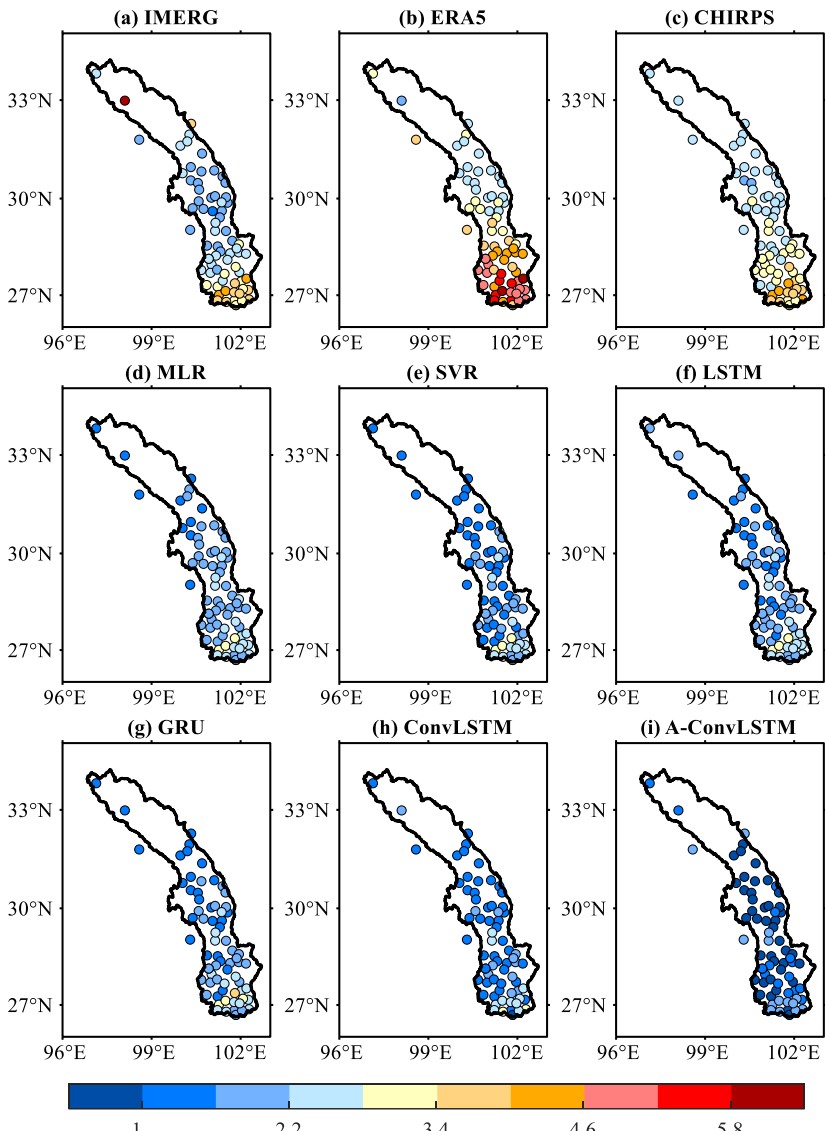

**Figure 10.** Average absolute error (*MAE*) for different models.

Aiming at overcoming the shortcomings of the original precipitation products, multi-source precipitation fusion models improve the accuracy of precipitation products by merging precipitation product data, ground observation precipitation and other auxiliary variable information. Because a higher *CC* value and lower *RMSE* and *MAE* values mean that models have a better performance, the redder points in Figure 8 and the bluer points in Figures 9 and 10 represent better results for the model. It can be seen from Figures 8–10 that each precipitation fusion model has a good effect on improving merged precipitation accuracy, especially downstream. It indicates that the addition of environmental variables such as wind speed, temperature, vegetation data and terrain factors can improve the accuracy of the precipitation products; this is similar to the conclusions of previous studies [39–43]. Due to varying model complexity and ability to handle nonlinear relationships, LSTM and GRU are comparable in precipitation fusion and slightly better than SVM and MLR. However, these models, which can only represent the temporal correlation between input factors and the response variable, need to expand the two-dimensional spatiotemporal vectors into one-dimensional vectors for their input and calculation, which inevitably results in losing the spatial distribution of the input factors. Thus, their performance is inferior to ConvLSTM and A-ConvLSTM. A-ConvLSTM uses the attention mechanism to select beneficial input factors for the ConvLSTM inputs to improve the accuracy of merged

precipitation. Compared with ConvLSTM, A-ConvLSTM can improve the precipitation fusion effect of downstream stations, indicating that it has a better improvement effect on stations with heavy precipitations. However, it is noted that, because there are fewer ground stations, there is still room for a further improvement of A-ConvLSTM upstream. In conclusion, from the performance evaluation based on the observed precipitation data, A-ConvLSTM has the smallest error and strong robustness and can better reflect the spatial distribution characteristics of precipitation.

### 3.2. Fusion Model Performance under Different Precipitation Intensities

The error between merged precipitation and observed precipitation may be different at different precipitation intensities. A quantitative and qualitative evaluation of errors under different precipitation intensities can reflect the performance of the precipitation fusion model more comprehensively. According to the grade standard of valley area precipitation, light precipitation, moderate precipitation and heavy precipitation refer to 24-h station precipitations of 0.1~10 mm/d, 10~25 mm/d and 25~50 mm/d, respectively. In this study, the total number of days with light precipitation, moderate precipitation, heavy precipitation and above heavy precipitation at all stations are 54,911, 16,917, 4585 and 728, respectively. Table 3 shows the errors of the original precipitation products and merged precipitation at all stations at different precipitation intensities.

**Table 3.** Accuracy of original precipitation products and merged precipitation at different precipitation intensities.

| Precipitation (mm/Day) | Indices | IMERG | ERA5 | CHIRPS | A-ConvLSTM |
|---|---|---|---|---|---|
| light precipitation (0.1~10) | CC | 0.182 | 0.161 | 0.167 | 0.509 |
| | RMSE (mm/d) | 4.85 | 7.03 | 4.4 | 3.19 |
| | MAE (mm/d) | 3.01 | 4.75 | 3.23 | 2.01 |
| | CSI | 0.320 | 0.237 | 0.277 | 0.427 |
| moderate precipitation (10~25) | CC | 0.171 | 0.093 | 0.091 | 0.508 |
| | RMSE (mm/d) | 9.83 | 8.73 | 10.61 | 6.05 |
| | MAE (mm/d) | 8.54 | 7.15 | 9.33 | 4.15 |
| | CSI | 0.137 | 0.158 | 0.113 | 0.514 |
| heavy precipitation (25~50) | CC | 0.098 | 0.012 | 0.007 | 0.356 |
| | RMSE (mm/d) | 24.53 | 23.08 | 26.78 | 13.33 |
| | MAE (mm/d) | 23.19 | 21.44 | 25.60 | 8.87 |
| | CSI | 0.015 | 0.025 | 0.001 | 0.485 |
| above heavy precipitation (>50) | CC | −0.026 | −0.101 | −0.143 | 0.109 |
| | RMSE (mm/d) | 57.75 | 56.19 | 60.70 | 37.60 |
| | MAE (mm/d) | 54.54 | 52.67 | 57.68 | 25.95 |
| | CSI | 0.001 | 0.001 | 0.000 | 0.367 |
| total (>0.1) | CC | 0.363 | 0.241 | 0.236 | 0.807 |
| | RMSE (mm/d) | 10.28 | 10.76 | 10.90 | 6.34 |
| | MAE (mm/d) | 5.97 | 6.80 | 6.53 | 3.18 |
| | CSI | 0.490 | 0.411 | 0.442 | 0.603 |

According to the changes in the indices from Table 3, it can be found that the accuracy of the three original precipitation products decreases with the increase in precipitation intensity. When the precipitation intensity is light, IMERG has the highest accuracy and the best precipitation detection performance. However, ERA5 has a large error, which may be because it tends to overestimate weak precipitation [1]. At an intensity of 10~50 mm/day, IMERG's *CC* is higher than for ERA5, while its *RMSE* and *MAE* are higher than for ERA5 and its *CSI* is lower than for ERA5. This indicates that ERA5 is better than IMERG in detecting moderate or heavy precipitation. In addition, it is worth noting that when the precipitation intensity is more than 25 mm/day, CHIRPS's *CC* and *CSI* are both less than 0.1. This indicates that CHIRPS has a weak ability to capture heavy precipitation, which

is similar to the conclusion of Bai et al. [71]. When the precipitation intensity exceeds 50 mm/d, all precipitation products' *CC*s are less than 0, the *CSI*s are less than 0.01, and the *RMSE*s and *MAE*s are more than 50 mm/d, which indicates that original precipitation products are less capable of capturing rainstorm in the Yalong River. This greatly limits their ability to detect extreme precipitation. From the comparison of indices with a precipitation intensity of more than 0.1 mm/day, the overall accuracy of IMERG is slightly better than that of ERA5 and CHIRPS.

After multi-source precipitation fusion by A-ConvLSTM, the correlation coefficient of the merged precipitation is increased, the error is reduced and the probability of successful detection is improved under different precipitation intensities. For example, compared with IMERG, *MAE* decreases by 1, 4.39, 14.32 and 28.6, and *CSI* increases by 0.107, 0.377, 0.469 and 0.366 when the precipitation intensities are light precipitation, moderate precipitation, heavy precipitation and above. When the precipitation intensity exceeds 50 mm/day, the merged precipitation's *CC* increases from −0.026 to 0.109, its *CSI* increases from 0.001 to 0.367, its *RMSE* decreases from 56.19 to 37.6, and its *MAE* decreases from 52.67 to 25.95. Therefore, merged precipitation has a certain ability to detect extreme precipitation. On the whole, it can be concluded that the A-ConvLSTM precipitation fusion model can correct and improve the original precipitation product data, and merged precipitation can reflect the intensity and spatial distribution characteristics of regional precipitation more accurately.

### 3.3. Application of Merged Precipitation in Streamflow Simulation

Precipitation is an important component of the water cycle and plays a critical role in streamflow simulations. Most of the existing hydrological models describe the process of precipitation forming a watershed streamflow through simulation methods. Thus, in this study, five main hydrological control stations, including Ganzi, Yajiang, Maidilong, Jinping and Tongzilin at the mainstream of the Yalong River, were used for streamflow simulation. Their daily average restored streamflow series from 2011 to 2020 were collected to evaluate the applicability of merged precipitation. Among them, 2011 was used as the warm-up period, 2012~2017 as the calibration period and 2018~2020 as the verification period. Table 4 shows the verification results of the hydrological models at each station.

**Table 4.** Accuracy of hydrological models at each station.

| Hydrologic Station | Hydrological Model | NSE | CC | MRE | RMSE (m³/s) |
|---|---|---|---|---|---|
| Ganzi | XAJ | 0.81 | 0.90 | 0.25 | 118.81 |
| | SWAT | 0.73 | 0.90 | 0.33 | 152.83 |
| Yajiang | XAJ | 0.94 | 0.98 | 0.12 | 175.43 |
| | SWAT | 0.85 | 0.94 | 0.15 | 301.92 |
| Maidilong | XAJ | 0.99 | 0.99 | 0.01 | 105.05 |
| | SWAT | 0.87 | 0.95 | 0.08 | 338.24 |
| Jinpin | XAJ | 0.94 | 0.97 | 0.11 | 267.72 |
| | SWAT | 0.85 | 0.94 | 0.17 | 301.92 |
| Tongzilin | XAJ | 0.94 | 0.97 | 0.10 | 410.77 |
| | SWAT | 0.85 | 0.93 | 0.18 | 664.22 |

It can be preliminarily determined from Table 4 that except for Ganzi Station, all models achieved good results in the streamflow simulation at each station, with an *NSE* of more than 0.85, a *CC* of more than 0.9, and an *MRE* of less than 0.2. Snowfall is an important component of precipitation in the upper Yalong River, resulting in a certain proportion of snowmelt production in the multi-year runoff [73]. Because the merged precipitation may have insufficient detection capability for snowfall and hydrological models cannot adequately reflect the process of snowmelt production, the runoff simulation error at Ganzi station is large. However, its *NSE*s are all above 0.7, its *CC*s are above 0.9 and its *RMSE*s are below 200, which indicates that the dispersion between simulated and observed streamflow is small. Thus, the simulated results of Ganzi station can still reflect the trend of the observed streamflow. In general, merged precipitation has good applicability in

streamflow simulation and can be used in hydrological forecasting to guide water resources management and allocation.

To visually evaluate and analyze the applicability of merged precipitation for streamflow simulation, the daily streamflow processes at five hydrological stations in the Yalong River are simulated by two hydrological models. The simulation results are shown in Figure 11.

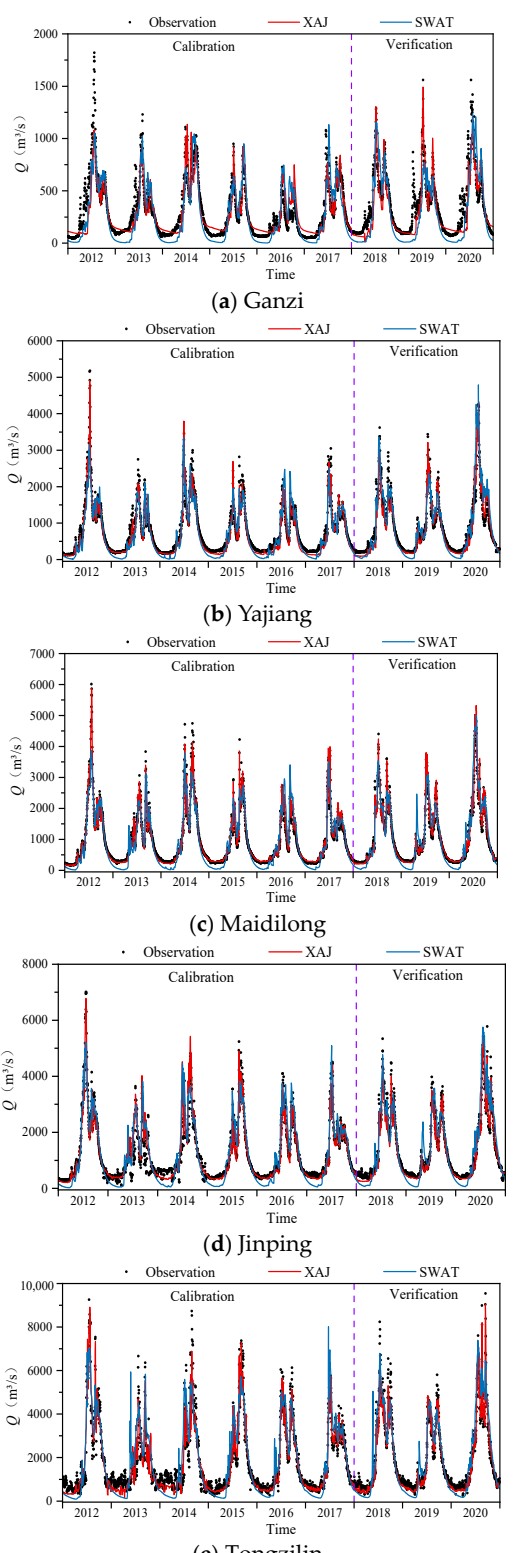

**Figure 11.** Daily streamflow simulation at five hydrological stations with two hydrological models.

As shown in Figure 11, the simulated streamflow's overall trends are consistent with the observed streamflow at each station, which highlights the main features of the observed streamflow. The simulated streamflow based on merged precipitation fits well with the observed streamflow peak, but there is an underestimation, especially at the Ganzi station. The flood characteristics of the Yalong River are characterized by a high peak, small volume and short duration. Thus, the accurate detection of the strong precipitation values helps to accurately simulate the flooding process. Though merged precipitation makes some corrections to the strong precipitation, it still underestimates the strong precipitation, leading to an underestimation of the streamflow peak. Especially in the upper Yalong River, the simulation is poor in some years at the Ganzi station because of fewer ground stations and the insufficient ability of merged precipitation to estimate snowfall, which affects the overall accuracy. However, the simulated streamflow can still represent the trend of the observed streamflow well and has the feasibility of an application.

In addition, it can be inferred from Table 4 and Figure 11 that the overall simulation results of XAJ are better than that of SWAT. This may be due to the complex structure of SWAT, which needs to consider many variables such as solar radiation, temperature and land use type, resulting in the superposition of various errors when the model parameters are optimized [74]. The XAJ model has a simple structure, requiring only precipitation and evaporation as input data, with less uncertainty in the inputs and parameters than SWAT. Nevertheless, as a kind of distributed hydrological model, SWAT can describe the water flow movement and its changing rules in a more detailed and comprehensive way based on physical equations, which represents the frontier of hydrological modeling research.

In summary, combined with the daily streamflow simulation evaluation indices and streamflow fitting diagram, it can be concluded that merged precipitation has good applicability in the Yalong River and can be applied to hydrological forecasting.

## 4. Conclusions

High-precision precipitation estimates with a high spatiotemporal resolution can effectively reflect the spatial and temporal distribution of a water cycle in a basin, which can guide scientific management and the allocation of water resources. This study focuses on using deep learning models to merge multi-source precipitation products with environmental variables to improve the spatiotemporal accuracy of precipitation in high-altitude areas. Thus, a multi-source precipitation spatiotemporal fusion model is proposed by coupling the attention mechanism and ConvLSTM, and the accuracy and applicability of the merged precipitation are evaluated from some perspectives. The main conclusions are as follows:

(1) The A-ConvLSTM model outperforms the comparison models (MLR, SVR, LSTM, GRU and ConvLSTM) and can effectively improve the precipitation estimation of the three precipitation products in time and space. Meanwhile, the merged precipitation correlates well with the ground observation precipitation with high consistency.

(2) The A-ConvLSTM model can correct and improve the original precipitation product data under different precipitation intensities and enhance the ability to detect strong precipitation but still underestimate its value.

(3) The streamflow simulation validation based on XAJ and SWAT models shows that the merged precipitation obtained by A-ConvLSTM can be applied to hydrological forecasting, indicating that merged precipitation has the potential to be extended for application in areas with no or little data.

This study proposes a useful method to improve the spatiotemporal accuracy of precipitation in high-altitude areas by exploring the spatiotemporal correlation between the original precipitation products and ground observation precipitation. However, it is undeniable that there is still room to improve the accuracy of merged precipitation, especially the detection of high-intensity precipitation, due to the small number of ground precipitation monitoring stations caused by the topography. With the further increase in ground stations in the future, the proposed model can be validated and applied to different regions.

**Author Contributions:** All authors contributed significantly to this manuscript. W.F.: writing—original draft preparation, software.; H.Q.: writing—review and editing, resources, funding acquisition.; G.L.: methodology.; X.Y.: investigation.; Z.X.: supervision. B.J.: data curation; Q.Z.: drawings. All authors have read and agreed to the published version of the manuscript.

**Funding:** This study was supported by the National Key Research and Development Program of China (2021YFC3200303) and the National Natural Science Foundation of China (52039004, 51979113).

**Acknowledgments:** The authors would like to thank Yalong River Hydropower Development Co., Ltd., China Meteorological DataService Center, the National Climate Center of China Meteorological Administration, NASA, JAXA, USGS, the University of California, Santa Barbara and ECMWF for providing the basic data.

**Conflicts of Interest:** The authors declare no conflict of interest.

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
