# Peer review of "A Method for Spatiotemporally Merging Multi-Source Precipitation Based on Deep Learning"

_remotesensing, doi:10.3390/rs15174160_

Round 1

Reviewer 1 Report

The paper "A method for spatiotemporally merging multi-source precipitation based on deep learning" presents the A-ConvLSTM method, which achieves significant improvement in merging multi-source precipitation and demonstrates good performance. The paper has a clear structure, reasonable experimental design, and detailed result analysis, providing valuable insights for readers. However, there are two minor issues that need clarification from the authors.

 In Figure 1, why are there blank areas in the DEM data around the river? Please provide an explanation.

On page 254, since the spatial interpolation is performed at a resolution of 0.01°, why are higher-resolution products not used for surface environmental variables such as NDVI? It seems that interpolating data with a resolution of 0.05° to 0.01° would not generate new information but only increase the number of grid cells. Additionally, what interpolation method was used by the authors?  Alternatively, if it was a typographical error and the resolution was actually interpolated to 0.1° according to the content in the figure, how did the authors handle such a large change in spatial resolution when downsampling DEM from 90 meters to approximately 10 kilometers (0.1°)?

Reviewer 2 Report

This study provides a novel method to merging multi-source precipitation data using attention mechanism and convLSTM. This method was demonstrated to be more accuracy than reanalysis dataset ERA5, satellite dataset, even for the different precipitation intensity. The result is interesting and meaningful. Therefore, I think this manuscript is well organized and written, but it needs to have a minor revision before acceptance.

Comments:

1.     Line 136-139: “It can be seen that the emergence of deep learning models can provide the possibility to handle the temporal-spatial nonlinear relationship between ground-observed precipitation and other precipitation products better.” Here, what does the nonlinear relationship between ground-observed precipitation and other precipitation products refer to? How does it form? Please give more explanation.

2.     Line 141-143: “In addition, most studies were developed in areas with dense precipitation stations and could not evaluate the practicability of the results from the perspective of rainfall-runoff relation well”. I did not understand the logic of his sentence. Why could it not evaluate the practicability of the results from the perspective of rainfall-runoff relation?

3.     Line 147-149: “(2) develop multi-source precipitation fusion to verify the effectiveness of the proposed model by comparing other models;” I think this point 2 is too much similar to point 1. I think it should be corresponding the section 3.2, so it should refer to verify the effective of the fusion model in the different precipitation intensity.

4.     Figure 1: the symbol of precipitation station is not clear, please replace it by another symbol. Besides, please explain the plus and minus in the upper left corner.

5.     Line 413-414: “However, increasing the number of sub-grids too much will cause too much useless information to interfere with the accuracy.” This sentence is contradicted to the sentence in line 409-410: “This may be because ConvLSTM and A-ConvLSTM can capture the spatiotemporal feature of precipitation and can comprehensively merge subgrid information.” In my view, A-ConvLSTM could get a more accurate weight. When the region increases, the weights should remain its accuracy by decreasing the weights of useless information. But as shown in figure 7a, when n increase from 5 to 15, the CC has a significant decrease, which is contract to the theory of ConvLSTM. Please give more explanation.

6.     Table 3, the word of “GPM” and “IMERG” are not consist in the manuscript, please keep consistency.

7.     Table 3, why the CC in the column of GPM, ERA5 and CHIRPS is so low? For moderate precipitation, CC of CHIRPS is 0.091, which indicating that there is no relationship between CHIRPS and ground-observation. Is it caused by the classification of precipitation intensity? I know that there is a bias of climatology of precipitation in ERA5 compared to ground-observation. Therefore, I suggest repeating this analysis using the percentile to classify the precipitation.

8.     Line 600-602: “XAJ calculates the total runoff from precipitation based on runoff the principle of generation under saturated conditions, which only requires precipitation and evaporation as input data, and thus has an advantage in the error accumulation problem.” Please rephrase this sentence to make it clear.

The quality of English Language is good, except for few sentence. I have given one sentence need be improved in my comments.

Reviewer 3 Report

General comment:

The study presents the evaluation of several gridded precipitation products and machine learning-predicted (multi-source precipitation fusion) precipitation in the Yalong River Basin (YRB). The manuscript is well-presented, but still lacks in several areas that require improvement and modification:

Introduction: While authors provided a comprehensive review of the literature; the introduction is still missing the relevant previous studies in the YRB and the reason for the need for this study in the YRB. The proposed multi-source precipitation fusion model could be evaluated in other areas as well.

Methods: add the description of all machine learning algorithms used in this study (only reported in the results section)

Results: Authors report that the proposed method provides a better representation of spatial and temporal patterns of precipitation; however, no temporal data are compared or reported. Daily precipitation plots are necessary to verify the authors claim. In several instances, “significance” is used but never mentions what statistical test was applied and/or the confidence level. Also, the streamflow comparison from two hydrological models does not show the performance of the proposed method with other methods/datasets.

The authors have evaluated the proposed method at meteorological locations only, which is reasonable for evaluation purposes. But what if the proposed method is to be applied in areas where ground observations are not available? I would suggest adding maps of currently available gridded precipitation datasets and the map from the proposed method for the YRB. This would add to the strength and findings of the study.

Specific comments:

Lines 23-24: report the units of RMSE and MAE

Line 35: accuracy or precision? Maybe, Accurate precipitation data can …..

Line 157: Yalong River Basin…

Lines 176-180: Looks like these sentences are not related to this study

Figure 1: precipitation stations are not visible, please use the different symbol

Table 1: station is not data? Please revise the table to report the data used from these stations?

Lines 200-2001: how the accuracy and completeness were significantly improved?

Lines 241-243: explain the hydrological effects? Compared with streamflow observations?

Line 380: CSI not reported before…please add full form

Lines 395-403: these are not results, suggest moving to methods section

Table 2: are these results for 20% test data? Also, add units of RMSE and MAE

Lines: 495-500: these must be in the methods section

Lines 538-554: also this in methods section

Section 3.3: this section is not providing the strengths of the new precipitation dataset compared to the other precipitation dataset. Probably remove this section or include the streamflow prediction from hydrological models using other precipitation datasets as well.

Lines 596-600: True. The accurate precipitation data does not determine that streamflow will be accurate. Thus, this analysis does not verify the authors claim of accurate precipitation data.

Round 2

Reviewer 3 Report

The authors have responded or addressed to several comments and suggestions. To provide more spatial and temporal information from the proposed approach, comparison of precipitation maps (sample grids) and time series plots are recommended.

Author Response

Point 1: The authors have responded or addressed to several comments and suggestions. To provide more spatial and temporal information from the proposed approach, comparison of precipitation maps (sample grids) and time series plots are recommended.

Response 1: Thank you so much for your selfless work and valuable comments and suggestions. The aim of this study is to propose a novel multi-source precipitation spatiotemporal fusion method for improving the spatiotemporal accuracy of precipitation. Based on the precipitation observed at the ground station, the CC, RMSE and MAE of each station under different precipitation are calculated to obtain the distribution of the stations, as shown in Figures 8 to 10. Meanwhile, in Tables 2 and 3, different precipitation results have been systematically compared and the advantages of A-ConvLSTM fusion precipitation have been analyzed. They have shown the temporal and spatial advantages of the proposed method in general. Please expert reviewer reviews the revised manuscript.